# Effects of helminths and anthelmintic treatment on cardiometabolic diseases and risk factors: A systematic review

Khanh Pham[1,2]*, Anna Mertelsmann[1], Keith Mages[3], Justin R. Kingery[4], Humphrey D. Mazigo[5], Hyasinta Jaka[6,7], Fredrick Kalokola[6,8], John M. Changalucha[9], Saidi Kapiga[9], Robert N. Peck[2,8,9], Jennifer A. Downs[2,8]

1 Division of Infectious Diseases, Weill Cornell Medicine, New York, New York, United States of America,
2 Center for Global Health, Weill Cornell Medical College, New York, New York, United States of America,
3 Samuel J. Wood Library, Weill Cornell Medicine, New York, New York, United States of America,
4 Department of Medicine, University of Louisville, Louisville, Kentucky, United States of America,
5 Department of Parasitology, Catholic University of Health and Allied Sciences, Mwanza, Tanzania,
6 Department of Internal Medicine, Catholic University of Health and Allied Sciences, Mwanza, Tanzania,
7 Department of Internal Medicine, Mwanza College of Health and Allied Sciences, Mwanza, Tanzania,
8 Department of Medicine, Weill Bugando School of Medicine, Mwanza, Tanzania, 9 Mwanza Intervention Trials Unit, Mwanza, Tanzania

* khp9007@med.cornell.edu

**Data Availability Statement:** All relevant data are within the manuscript and its Supporting information files.

## Abstract

### Background

Globally, helminth infections and cardiometabolic diseases often overlap in populations and individuals. Neither the causal relationship between helminth infections and cardiometabolic diseases nor the effect of helminth eradication on cardiometabolic risk have been reviewed systematically in a large number of human and animal studies.

### Methods

We conducted a systematic review assessing the reported effects of helminth infections and anthelmintic treatment on the development and/or severity of cardiometabolic diseases and risk factors. The search was limited to the most prevalent human helminths worldwide. This study followed PRISMA guidelines and was registered prospectively in PROSPERO (CRD42021228610). Searches were performed on December 10, 2020 and rerun on March 2, 2022 using Ovid MEDLINE ALL (1946 to March 2, 2022), Web of Science, Cochrane Library, Global Index Medicus, and Ovid Embase (1974 to March 2, 2022). Randomized clinical trials, cohort, cross-sectional, case-control, and animal studies were included. Two reviewers performed screening independently.

### Results

Eighty-four animal and human studies were included in the final analysis. Most studies reported on lipids (45), metabolic syndrome (38), and diabetes (30), with fewer on blood pressure (18), atherosclerotic cardiovascular disease (11), high-sensitivity C-reactive protein (hsCRP, 5), and non-atherosclerotic cardiovascular disease (4). Fifteen different

**Funding:** The study received support from NewYork-Presbyterian Hospital (NYPH) and Weill Cornell Medical College (WCMC), including the Clinical and Translational Science Center (CTSC) (UL1 TR000457) and Joint Clinical Trials Office (JCTO), as well as the Weill Cornell T32 training grant (T32AI007613 Research Training in Infectious Diseases). KP is supported by the Burroughs Wellcome Fund/American Society of Tropical Medicine & Hygiene Postdoctoral Fellowship in Tropical Infectious Diseases. RP is supported by the National Heart, Lung, and Blood Institute (R01 HL160332). JD is supported by the National Institute of Allergy and Infectious Diseases (R01 AI 168306). The funders had no role in study design, data collection and analysis, decision to publish, or preparation of the manuscript.

**Competing interests:** The authors have declared that no competing interests exist.

helminth infections were represented. On average, helminth-infected participants had less dyslipidemia, metabolic syndrome, diabetes, and atherosclerotic cardiovascular disease. Eleven studies examined anthelmintic treatment, of which 9 (82%) reported post-treatment increases in dyslipidemia, metabolic syndrome, and diabetes or glucose levels. Results from animal and human studies were generally consistent. No consistent effects of helminth infections on blood pressure, hsCRP, or cardiac function were reported except some trends towards association of schistosome infection with lower blood pressure. The vast majority of evidence linking helminth infections to lower cardiometabolic diseases was reported in those with schistosome infections.

## Conclusions

Helminth infections may offer protection against dyslipidemia, metabolic syndrome, diabetes, and atherosclerotic cardiovascular disease. This protection may lessen after anthelmintic treatment. Our findings highlight the need for mechanistic trials to determine the pathways linking helminth infections with cardiometabolic diseases. Such studies could have implications for helminth eradication campaigns and could generate new strategies to address the global challenge of cardiometabolic diseases.

## Author summary

Helminth infections are caused by parasitic worms and affect over 1.5 billion people worldwide. Helminth infections and cardiometabolic diseases are both common and overlap with one another in many parts of the world. Studies have separately examined the relationship between helminth infections and various cardiometabolic diseases, but the relationships overall, as well as the impact of treatment of parasitic worms, have not been studied systematically. The authors conducted a systematic review to assess the impact of helminth infections, and treatment of helminth infections, on cardiometabolic diseases and risk factors. Eighty-four total studies were analyzed and included in the final review. People and animals infected with helminths were generally found to have fewer cardiometabolic disease risk factors including better overall cholesterol profiles, less diabetes, and less atherosclerotic heart disease than uninfected study participants. After treatment of helminth infections, participants frequently experienced worsening in those cardiometabolic measurements. There were no consistent effects of helminth infections on blood pressure, high-sensitivity CRP (an inflammatory marker), or other cardiac function. In summary, helminth infections may offer protection against certain cardiometabolic diseases and risk factors. More studies are needed to elucidate the pathways linking helminth infections with cardiometabolic diseases as it may impact how we treat both disease processes in regions where both are prevalent.

## Introduction

Cardiometabolic diseases, which include cardiovascular disease and diabetes, are the leading cause of death worldwide, accounting for 32% of global deaths [1]. The incidence of cardiometabolic diseases and well-recognized traditional risk factors for these diseases, such as hypertension and dyslipidemia, have been rising particularly in low- and middle-income countries

(LMICs) [2]. Novel interactions with the environment may play a critical role in the development of cardiometabolic diseases and risk factors in these regions.

Helminths infect approximately one-quarter of the world's population [3] and account for numerous neglected tropical diseases, causing significant morbidity and mortality mainly in LMICs. Given that helminths have co-existed with humans for millennia, it would not be surprising if coevolution has resulted in beneficial effects of helminth infections on human health. Previous systematic reviews have reported that helminth infections can be associated with favorable metabolic function and outcomes, but other cardiometabolic measures and the effects of anthelmintic treatment were not included [4,5]. Additionally, other studies have demonstrated inconsistency in the literature, with some showing that helminth infections worsened or had no effect on metabolic measures [6–8]. Against the backdrop of the World Health Organization's efforts to eliminate helminth infections by 2030 [9] and these previous data, we sought systematically to review data to investigate: (1) associations between helminth infections and cardiometabolic diseases and risk factors, and (2) whether helminth eradication could alter the prevalence and incidence of cardiometabolic diseases and risk factors. Given a growing number of studies reporting relationships between helminths and cardiovascular health, we hypothesized that helminth infections would be associated with fewer cardiometabolic diseases and risk factors, and that anthelmintic treatment would increase the risk or development of these noncommunicable diseases and conditions.

## Methods

We conducted a systematic review to assess the effects of helminth infections on the development and/or severity of cardiometabolic diseases and risk factors. This study was reported in accordance with the Preferred Reporting Items for Systematic Reviews and Meta-Analyses (PRISMA) guidelines and was registered prospectively in PROSPERO (CRD42021228610).

### Data sources and searches

A comprehensive search was developed and performed on December 10, 2020 and rerun on March 2, 2022 using Ovid MEDLINE ALL (1946 to March 2, 2022), Web of Science, Cochrane Library, Global Index Medicus, and Ovid Embase (1974 to March 2, 2022). Both English and foreign language papers were included. Databases were searched from inception. Authors were not contacted for further information. Studies were uploaded to the Covidence platform (Melbourne, Australia) for conduct of the screening and extraction phases.

Included study types were randomized clinical trials, cohort, cross-sectional, and case-control studies. Both human and animal studies were included. S1 Table summarizes our study's predefined search terms, keywords, and study types included, with the full search strategy detailed in S1 Appendix.

Predefined search terms for cardiometabolic diseases and risk factors included systolic blood pressure, total cholesterol, high-density lipoprotein cholesterol (HDL), high-sensitivity C-reactive protein (hsCRP), coronary artery disease (CAD, which included myocardial infarction (MI) and large vessel atherosclerosis), cerebrovascular disease, peripheral vascular disease, heart failure, rheumatic heart disease, congenital heart disease, and cardiomyopathies. These terms were generated by identifying modifiable biomedical components included in 3 widely used calculated cardiac risk equations (Atherosclerotic Cardiovascular Disease 2013 (ASCVD 2013) Risk Calculator, Reynolds Risk Score, and the Framingham Risk Score with Lipids), as well as by including terms in the World Health Organization's definition of cardiovascular disease [1].

Predefined search terms for metabolic syndrome included abdominal obesity, atherogenic dyslipidemia, raised blood pressure, insulin resistance, and glucose intolerance. These terms were generated according to the biomedical components that characterize metabolic syndrome, as defined by the National Cholesterol Education Program—Adult Treatment Panel (ATP) III report [10]. Prothrombotic and proinflammatory states, also components of the ATP III definition, were not included in the search as there is unclear guidance on how to apply these risk factors to clinical patient care [10]. Additionally, articles that reported host factors without any clear correlation to cardiometabolic diseases were excluded.

Predefined search terms for helminths pathogenic in humans included the soil-transmitted nematodes, filarial nematodes, platyhelminth flukes, and the platyhelminth tapeworm, *Taenia solium*, as shown in S1 Table. Corresponding diseases for these etiologic agents were also developed into the search (e.g., schistosomiasis for *Schistosoma* worm species). These helminths were chosen because they are the most prevalent human helminthiases worldwide [11].

## Eligibility criteria and study selection

Studies were eligible if they fulfilled all of the following: 1) were a correct study type; 2) had a correct parasite genus and species; 3) had a correct cardiometabolic disease or risk factor; 4) described helminth infection as the exposure and cardiometabolic disease or risk factor as the outcome of the study, or vice versa; and 5) had available full texts or abstracts with enough information for further assessment. Excluded papers included review papers without original data and case reports or case series. Duplicate records were removed. Two investigators independently screened titles and abstracts to identify studies eligible for inclusion using the above criteria, and a third investigator independently resolved discrepancies.

## Data extraction and quality assessment

After the completion of the screening methods, data were extracted and quality assessment was performed. Data extracted included helminth species and study type; country of origin; sample size; cardiometabolic outcomes; key findings; effects of anthelmintic therapy on outcomes, if applicable; and study limitations. Quality assessment of data was performed using the Downs and Black checklist [12]. Studies were categorized as "poor," "fair," "good," or "excellent" based on a summative score derived from this checklist, and only those of "fair" quality or above were included in the final analysis.

## Data synthesis and analysis

Data were synthesized and analyzed in Stata MP/Version 17 (College Station, Texas) and Microsoft Excel Version 16.53. Descriptive statistics were applied, using medians for continuous variables and proportions for categorical variables.

Cardiometabolic diseases and risk factors reported from included articles were categorized for analysis. These 7 categories included lipids, metabolic syndrome and related parameters, diabetes and related parameters, blood pressure and other cardiovascular hemodynamics, cardiovascular disease (which included CAD, MI, and large vessel atherosclerosis), hsCRP, and non-atherosclerotic cardiovascular disease. It was anticipated that some studies would report on more than one cardiometabolic outcome within the same study; if this occurred, then each outcome was grouped accordingly under the corresponding category (e.g., if study A reported on both lipids and diabetes, then we categorized study A under both the lipids and diabetes categories, highlighting the specific outcome under each corresponding category and

incorporating study A in the analysis of each category). Because some components of metabolic syndrome overlap with other cardiometabolic categories, the metabolic syndrome category only included anthropometric measurements (e.g., body mass index (BMI), waist circumference (WC), and waist-to-hip ratio (WHR)), glucose intolerance, and insulin resistance. Other components of metabolic syndrome, such as blood pressure or dyslipidemia, were grouped into the blood pressure and lipids categories, respectively, and not included in the metabolic syndrome category.

Due to the heterogeneity of study designs and inclusion of different cardiometabolic measures and helminth infections, a meta-analysis was not possible.

## Results

Titles and abstracts of 8,646 studies were screened, after which 276 full-text papers were assessed for inclusion. A total of 118 studies then underwent data extraction and quality assessment with 84, which were determined to be of fair quality or higher, included in the final analysis (Fig 1). Fifteen different helminth species were represented, and these included *Ancylostoma duodenale*, *Ascaris lumbricoides*, *Brugia malayi*, *Clonorchis sinensis*, *Fasciola hepatica*, *Necator americanus*, *Onchocerca volvulus*, *Opisthorchis viverinni*, *Schistosoma* species (including *S. haematobium*, *japonicum*, and *mansoni*), *Strongyloides stercoralifigs*, *Taenia solium*, *Trichuris trichiura*, and *Wuchereria bancrofti*. *Schistosoma* species were most represented (*S. mansoni* (26), *S. japonicum* (8), unspecified *Schistosoma* species (7), and *S. haematobium* (2)) and studied in 43 studies, followed by mixed helminths (12), *Strongyloides stercoralis* (9), and others (20). The 34 records that did not meet Downs and Black criteria of "fair" or better quality are reported in S1 Supplemental Table. The associations of helminths with each of the 7 categories of cardiometabolic diseases and risk factors are discussed in the following sections.

### Helminths and lipid profiles

Forty-five studies examined helminth infections and serum lipid profile, including total cholesterol, low-density lipoprotein cholesterol (LDL), HDL, and/or triglycerides (S2 Table). These studies comprised 27 human and 17 animal studies, and 1 mixed human/animal study.

**Studies examining serum lipids before and after anthelmintic treatment.**   Of the 45 studies, 6 reported on serum lipids both before and after anthelmintic treatment. Three of these studies were conducted in humans [8,13,14] and 3 were conducted in animal models [15–17].

**Human studies examining serum lipids before and after anthelmintic treatment.**   One cluster-randomized trial of 1898 adults in Uganda reported lower total cholesterol and LDL at baseline in those with *S. mansoni* infection than those without [13]. The authors observed a dose-response relationship in which those with moderate to heavy *S. mansoni* infection had the lowest LDL and triglycerides. Similarly, individuals infected with *S. stercoralis* also had lower LDL than those without *Strongyloides* infection. In clusters that received community-wide intensive anthelminthic treatment with quarterly single-dose praziquantel and triple-dose albendazole, the mean LDL increased after 4 years of follow-up, as compared to the mean LDL in communities that received single-dose anthelmintic treatment, which was a significant trend (2.86 vs 2.60 mmol/L; adjusted mean difference, 0.26 [95% CI, −0.03 to 0.56]; P = .08). While this study provides robust evidence for LDL, it should also be noted that ongoing helminth transmission in both study arms at the end of the study and a cluster-randomized design with data reported as community means limited the study's ability to investigate outcomes at an individual level.

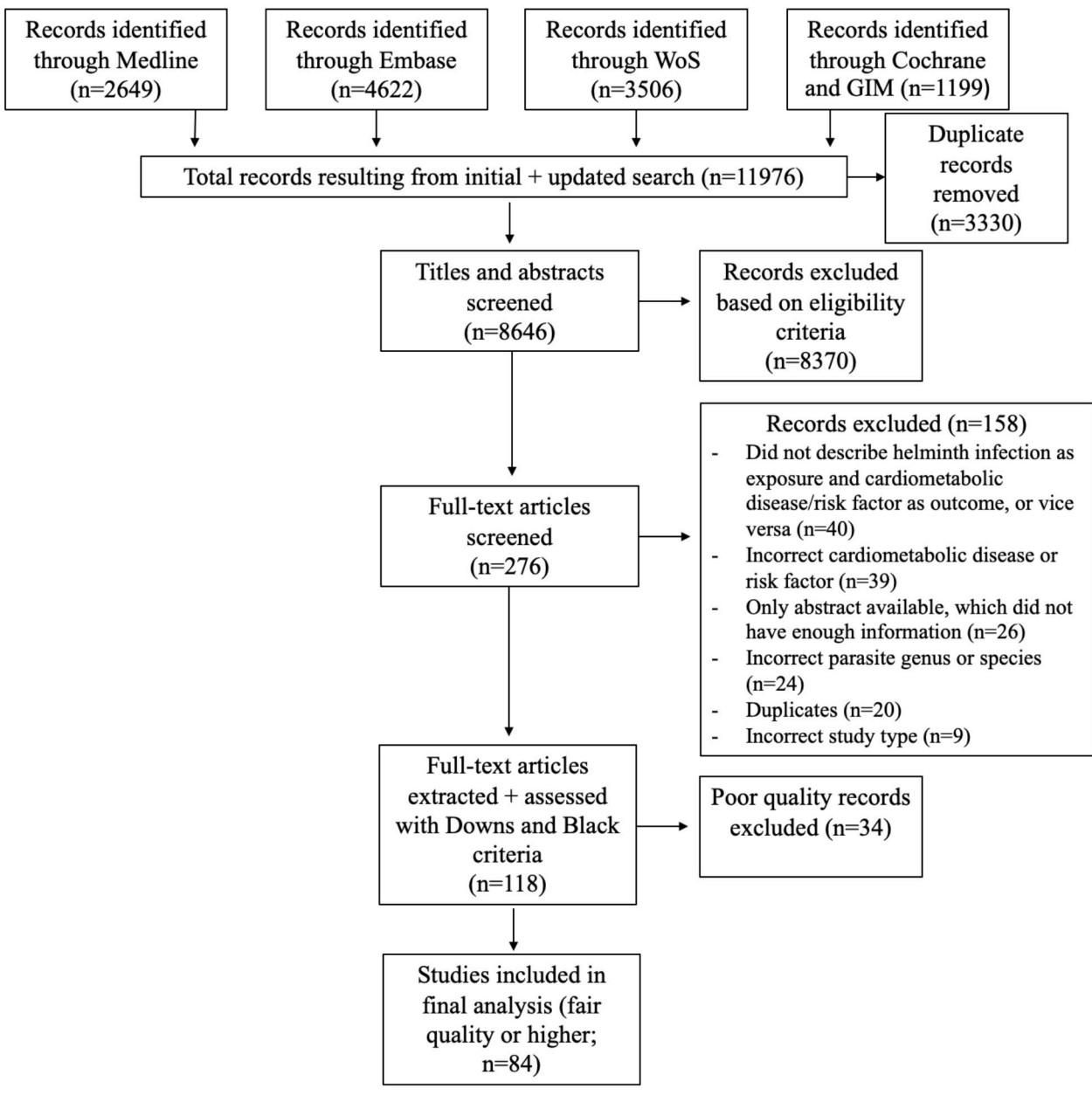

**Fig 1. PRISMA flow diagram of systematic review.**

A second human study reported significantly elevated HDL but no differences in serum total cholesterol, LDL, or triglycerides at baseline in 200 adults with *O. viverrini* infection versus 200 uninfected adults [14]. Six months after treatment with praziquantel, a further increase in HDL was observed.

In contrast, a household, cluster-randomized controlled trial [8] investigated lipid profiles in 1669 Indonesian persons with mixed helminth infections and found no significant associations between helminth infection and serum lipids either at baseline or 52 weeks post-anthelmintic treatment with albendazole.

In summary, 2 out of 3 human studies found that alterations in serum lipids were associated with helminth infections. Treatment of infection led to an increase in serum lipids in both studies [8,9].

**Animal studies examining serum lipids before and after anthelmintic treatment.**   One mouse study examined lipid changes in male mice exposed to *S. japonicum* cercariae or soluble egg antigen (SEA) [15]. They reported no differences in lipid profiles between cercariae-exposed and unexposed mice at baseline, but mice that were effectively treated with praziquantel had developed elevated LDL and HDL levels by 9 weeks post-treatment, compared to uninfected or persistently infected mice.

One sheep study found lower LDL in *Fasciola*-infected animals at baseline that increased 28 days after anthelmintic treatment with triclabendazole and levamisole [16]. This was also accompanied by newly elevated total cholesterol and HDL after treatment of infection. This study also reported decreased serum triglycerides after anthelmintic treatment. Limitations included studying only 30 sheep of unclear sex distribution, and possible confounding effect of a peroxisome proliferator receptor alpha agonist, which can affect lipids and was included with the treatment of fasciolosis. Additionally, although infected sheep were treated and had a significant reduction in parasite eggs, sheep still had up to 100,000 eggs per gram of stool post-treatment, suggesting that infection was not completely eradicated.

In another animal study using *F. hepatica*-infected sheep, the serum total cholesterol, LDL, HDL, and triglycerides were all lower at baseline and remained low with only minor, non-significant increases on day 28 after treatment of infection with triclabendazole and levamisole when compared to uninfected sheep [17]. However, by day 56 post-treatment, differences in total cholesterol, LDL, and triglycerides were no longer present, supporting the hypothesis that anthelmintic treatment would remove the beneficial effects of helminth infection on cardiometabolic risk factors. Limitations of this study included small sample size of 25 sheep and that the study appeared to only compare pre-post treatment lipids in the infected group to baseline measurements obtained in the control group.

In summary, 2 out of 3 animal studies demonstrated that helminth infection was associated with lower lipid levels [16,17]. All 3 studies showed that anthelmintic treatment subsequently increased lipid levels in these animal models [15–17].

**Summary of studies examining serum lipids before and after anthelmintic treatment.** Taken together, 4 of these 6 studies (67%, 2 human, 2 animal) reported that helminth infection was associated with beneficial lipid levels at baseline [13,14,16,17], and the majority (5 of 6 studies, or 83%) demonstrated some increases in various serum lipids after anthelmintic therapy [13–17]. Lower LDL was the most frequently reported lipid measurement associated with helminth infections among these studies at baseline, with higher HDL, lower triglycerides, and lower total cholesterol reported less frequently. It should also be noted that HDL was consistently reported to increase after infected individuals were treated [14–16], which in practice could translate to a clinical benefit given that cardiovascular risk is known to increase sharply as HDL levels fall below 40 mg/dL [18] and that the risk of CAD increases by 13% for every 10% reduction in HDL [19]. This suggests that helminth infections and anthelmintic treatment may preferentially affect certain types of lipids through different mechanisms that warrant further investigation. Interestingly, these effects were observed predominantly in helminths that affect the liver, suggesting intra-hepatic processing of cholesterol may be altered by these liver flukes. No significant trends were noted when these studies with pre-post anthelmintic treatment data were stratified by age or sex.

**Cross-sectional associations of helminth infections with serum lipids.**   Of the 45 studies cross-sectionally comparing serum lipids in human and animal studies, 36 reported lower lipids in association with helminth infection, 23 demonstrated no relationship, and 16 showed

that helminth infection was associated with higher lipids. Given heterogeneity in lipid trends within many of the studies, each lipid measurement will be discussed separately with comparisons between human versus animal studies.

Forty-three of these studies reported on total cholesterol [6–8,13–17,20–55]. Of these, 28 reported on participants infected with *Schistosoma* species or with mixed helminths that included *Schistosoma* species [6,7,13,15,25–28,30,31,35,37,39–44,46,48–55]. These 28 studies mostly reported that infected individuals or those challenged with helminth products had lower total cholesterol than uninfected participants. Including all helminths in other studies, 96% (24 of 25 total studies, which consisted of 11 human and 13 animal studies with 1 mixed study) that reported a relationship between helminth infection and lower total cholesterol found this inverse association to be significant [13,17,24–31,33,35,38–41,44,46–51,54,55]. In contrast, 6 studies (14%, all with different helminths with 4 human and 2 animal studies) reported associations with increased total cholesterol [6,20,21,33,34,36], and 13 (30%, 10 human and 3 animal studies, of which 6 examined schistosome infections) demonstrated no effect of infection on total cholesterol [7,8,14–16,22,23,32,36,37,42,43,53]. Notably, some studies reported differences in total cholesterol depending on factors such as duration of infection, co-morbidities, type of helminth infection, route of exposure to infection, geographic locations (e.g., urban vs rural or endemic vs. non-endemic), and BMI. For example, rabbits infected with *C. sinensis* developed higher total cholesterol during the first week of infection which then decreased below baseline levels during weeks 2–8 of infection [33]. In contrast, *S. mansoni*-infected mice had significantly lower total cholesterol than uninfected mice after exposure to schistosome eggs or egg antigens, but not to cercariae [44]. In another study, *S. haematobium* infection was associated with lower total cholesterol and other lipids only in overweight and obese infected people [52]. Although such heterogeneity of potential confounders makes comparisons challenging, the majority of these studies (24 of 43, or 56%) found that helminth infections, mostly with schistosomes but including other helminths as well, were significantly associated with lower total cholesterol. These 24 studies included robust human (n = 10 with 1 mixed study) and animal (n = 13) studies with large sample sizes and diverse study designs and participants, suggesting that beneficial effects of helminth infections on total cholesterol may be generalizable.

Twenty-nine studies reported on LDL [6–8,13–17,22,25,27–30,34–36,39,42,43,45–48,50–54], and the majority of these [19] focused on those infected with *Schistosoma* species. Eighteen (10 human, 8 animal) of the 29 studies (62%) found infected participants to have lower LDL than uninfected participants [7,13,16,17,25,27–30,34,35,39,45–48,50,51,53], of which 11 were in those with schistosome infections. By comparison, 3 studies (10%; 2 human, 1 animal) reported higher LDL in those with different helminth infections [6,34,36], and 10 (34%; 8 human, 2 animal), including 6 on schistosomes, showed no effect [7,8,14,15,22,36,42,43,52,54]. Similar to studies of total cholesterol, differences in LDL also depended on additional factors in some studies. For instance, *S. stercoralis* was associated with lower LDL in people with alcoholism, but with higher LDL in people who were not alcoholics [34]. Similarly, *O. viverrini*-infected hamsters had higher LDL than uninfected hamsters during acute infection, but there were no differences in LDL between the two groups as the infected hamsters progressed to chronic infection [36]. In total, almost 2/3 of the studies that reported on LDL demonstrated lower LDL in infected versus uninfected participants. Similar to the observation with total cholesterol, there were strong human and animal studies representing a breadth of helminth infections and study designs that supported this trend.

More heterogeneity was observed for associations between helminth infection and HDL or triglycerides. Of the 31 studies investigating HDL [6–8,13–17,22,25,27–30,32,34–37,39,42,43,46–48,50–54,56], 13 (42%; 7 human, 6 animal) found lower HDL [6,17,27,29,36,37,39,46,48,50,52,54,56], 9 (29%; 7 human, 2 animal) reported higher HDL [14,25,28,30,34,42,43,51,53],

and 9 (29%; 6 human, 3 animal) saw no association between helminth infections and HDL levels [7,8,13,15,16,22,32,35,47]. Notably, fewer than 50% of these studies showed lower HDL levels associated with infection, in contrast to high frequencies of studies that identified such inverse relationships between helminths and total cholesterol or LDL. Given the known association between lower HDL levels and increased CAD, it should be noted that this inverse relationship suggests that helminth infections may translate clinically to lower cardioprotection with respect to HDL [18,19]. In fact, only 29% of the studies showed a potentially protective effect with higher HDL levels in those with helminth infections. As with the other lipid measurements, the majority of the studies reporting either lower or higher HDL levels focused on schistosome infections, while those showing no effect had an even split between mixed helminth and schistosome infections.

Lastly, 35 studies reported on triglyceride levels with 20 showing lower triglycerides (57%; 12 human, 7 animal, 1 mixed study) [13,17,25,26,28,30,34,37,39,41–43,46,48,50–54,56], 3 (9%; 2 human, 1 animal) reporting higher triglycerides [29,32,36], and 15 (43%; 8 human, 7 animal) demonstrating no association with helminth infections [6–8,14–16,22,27,34–36,40,47–49]. The majority of these studies focused on schistosome infections. None of the 3 studies showing higher triglyceride levels investigated schistosome infection. Some of these studies reported varying trends of triglycerides in their same cohort of participants, similar to that observed in the total cholesterol and LDL studies. For example, one study documented significantly lower triglycerides in *S. mansoni*-infected mice that were fed normal diet, though not in those fed a high-fat diet (HFD), when compared to uninfected controls [37]. In a human study of *S. mansoni* infection in two regions of different schistosome endemicity in Ethiopia, no differences in triglyceride levels were seen between infected and uninfected individuals within the same region, but infected individuals in the highly endemic region had lower triglycerides than uninfected individuals in the *S. mansoni*-nonendemic region [48], possibly due to undiagnosed infections in the highly endemic region. The majority of these studies (57%) found lower triglycerides in infected versus uninfected individuals.

**Summary**: Overall, 45 studies reported on serum lipids with a median sample size of 167.5 study participants. Most (36 of 45 studies) demonstrated that helminth infections were associated with lower serum lipid levels cross-sectionally. Of the 6 that reported on the effects of anthelmintic treatment, 5 (2 human and 3 animal studies) showed that post-treatment serum lipid levels increased.

Of note, 12 human studies [7,8,13,21,26,32,34,39,45,47,48,52] focused on individuals with a median age of less than 50 years (38.5 [IQR 31.2–44.1]) and a median of 50% [46.5–59.1%] women. Among these, 6/11 reported lower total cholesterol, 6/8 reported lower LDL, and 5/9 reported lower triglycerides in those with helminth infections. Only 1/9 studies observed increased HDL associated with infection. Together, these studies suggest that the beneficial impact of helminth infections on unhealthy lipid levels may be highest in adults less than 50 years of age. In contrast, among the 8 human studies [6,22,25,28,42,43,53,56] with median participant ages ≥ 50 years (65.4 [54.2–69.5]) and fewer females (median 26% [16.2–40.8%]), only 2/7 studies observed lower total cholesterol and 3/7 observed lower LDL with helminth infections. However, helminth infections were associated with higher HDL in 5/8 studies and lower triglycerides in 6/8. Conclusions in older adults are more limited due to fewer studies, and are particularly limited for older women, who were underrepresented in these data. These data suggest that older men may derive more cardioprotection from helminth infections than older women, and that the beneficial lipid effects may be derived from different mechanisms in younger versus older adults with helminth infections. More rigorous studies, including those that control for infection intensity when examining effects of age and sex and those that have older female participants, are needed.

## Helminths and metabolic syndrome

Thirty-eight studies reported on the associations of helminths with metabolic syndrome and related measures, such as BMI, WC, WHR, glucose intolerance, insulin level, and/or homeostatic model assessment for insulin resistance (HOMA-IR) (S3 Table). These studies comprised 24 human and 13 animal studies, and 1 mixed study.

**Studies examining metabolic syndrome before and after anthelmintic treatment.**   Ten of the 38 studies investigated effects of helminth infections on metabolic syndrome measures before and after anthelmintic treatment [8,13–17,57–60]. Seven (4 animal, 3 human) of these reported an increase in metabolic syndrome and related measures and 3 (all human) demonstrated no effect on these factors after anthelmintic treatment.

**Human studies examining metabolic syndrome before and after anthelmintic treatment.**   In humans, two studies documented lower insulin levels and but no difference in random blood glucose levels in *S. stercoralis*-infected individuals when compared to uninfected people at baseline in two different cohorts: infected diabetic vs. uninfected diabetic individuals in one study [58], and infected, non-diabetic obese vs. uninfected, non-diabetic obese individuals in another study [59]. In individuals who received treatment with single-dose albendazole and ivermectin, both insulin and random blood glucose levels increased at 6 months of follow-up compared to pretreatment levels in those with known diabetes, and only insulin levels increased in those receiving the same anthelmintic regimen and duration of follow-up but who were obese and non-diabetic.

In a double-blind, household-cluster-randomized, placebo-controlled clinical trial of people with and without mixed helminth infections, no differences in levels of insulin or glucose or HOMA-IR were observed at baseline, but those with helminth infections who were randomized to receive albendazole treatment experienced a significant increase in insulin resistance (HOMA-IR) at 52 weeks compared to those randomized to placebo treatment (estimated treatment effect, 0.031 [95% CI, 0.004–0.059]; P = 0.04) [8]. Three other human studies showed no effect of helminth infection or anthelmintic treatment on measures of metabolic syndrome [13,14,57]

In summary, 50% (3 of 6) of human studies demonstrated increases in measures of metabolic syndrome after treatment of helminth infection.

**Animal studies examining metabolic syndrome before and after anthelmintic treatment.**   Mouse models consistently showed negative associations of metabolic syndrome parameters with helminth infections. *S. japonicum*-infected male mice had lower body weight and less insulin resistance than uninfected mice at baseline. Nine weeks following treatment with praziquantel, body weight and HOMA-IR of previously-infected mice increased compared to chronically-infected, untreated mice [15]. In a second mouse model, *S. mansoni*-infected mice had significantly lower blood glucose than uninfected mice at baseline, which had increased 7 and 14 days after infected mice received praziquantel [60]. Two other animal studies, using sheep and investigating the effects of *Fasciola* infection, reported similar trends in blood glucose before and after anthelmintic treatment with triclabendazole and levamisole [16,17], as seen in the mouse studies.

In summary, all 4 animal studies showed inverse assocations (lower measures of metabolic syndrome) with helminth infection and a subsequent increase in these measures after infection was treated.

**Summary of studies examining metabolic syndrome before and after anthelmintic treatment.**   Taken together, the majority of studies (7 of 10 (70%), with 4 animal and 3 human studies) reported lower body weight, blood glucose, insulin level, and insulin resistance in association with helminth infections [8,15,17,58–60]. Subsequently, these parameters

seemed consistently to increase after treatment of infected study participants. The 3 studies demonstrating no effect of anthelmintic therapy on parameters of metabolic syndrome were all human studies and notably also did not show any baseline differences in metabolic syndrome between helminth-infected and uninfected people [13,14,57]. Of note, the human studies included in this section had median participant ages of < 50 years (39.3 [34–44.3]), with approximately equal distributions of men and women. Although few in number, these human studies mostly suggest that anthelmintic treatment may paradoxically predispose both men and women younger than 50 to an increased risk for metabolic syndrome.

**Cross-sectional associations of helminth infections with metabolic syndrome.** Thirty-eight studies examined the baseline associations between helminth infections on metabolic syndrome and related measures, of which 22 (58%; 12 animal, 9 human, 1 mixed human/animal) demonstrated lower metabolic syndrome and related measures [15–17,25,28,30,37,42,43, 47,48,51,56,58–66], 14 (37%; 13 human, 1 animal) showed no relationship [7,8,13,14,20,21,26, 32,45,52,53,57,67,68], and 3 (8%; 2 human, 1 animal) reported an increase or variable changes in metabolic syndrome and related measures in association with helminth infections [6,66,69]. A large proportion of these studies (21/38) reported on blood levels of glucose or insulin; some also measured glucose intolerance or insulin resistance. Some examined anthropometric measurements, such as BMI, WC, and WHR, as well as prevalence of metabolic syndrome.

Two studies in China focused on people with history of prior schistosome infection (PSI). The first showed a lower prevalence of obesity and abdominal obesity among 465 men with PSI (8.4% and 24.3%, respectively) when compared to a control group of 1132 never-infected men (16.4% and 41.8%, respectively) [43]. Women were also included in this study, but there were no differences in obesity or abdominal obesity when the smaller numbers of previously infected (n = 61) and uninfected (n = 284) women were compared to each other [43]. BMI was lower in both men and women, and fasting blood glucose was also lower in women with previous infection compared to their uninfected counterparts. In the second study of 1,597 men, the prevalence of metabolic syndrome and its components, including central obesity, hypertriglyceridemia, hypertension, and hyperglycemia, was lower in the group with PSI versus those never infected [42]. Although both studies included large sample sizes, women were underrepresented in the first study (17.8%) and not included at all in the second; both also included older individuals with an average age of ~65 years. Additionally, PSI was diagnosed as ultrasonographic liver abnormalities plus participants' recall of past exposure to contaminated water and anthelmintic treatment, but other liver diseases were not reported as being excluded.

Other animal studies and human studies from Brazil, China, Egypt, Ethiopia, the Netherlands, and the United States reported similar inverse relationships between active schistosome infection and measures of metabolic syndrome [15,25,28,30,37,48,51,56,60–62,70]. Of note, 7 other studies did not demonstrate these associations, with 4 (3 human, 1 animal) showing no association [26,52,53,67], 2 (1 human, 1 animal) reporting increased insulin resistance, glucose intolerance, or insulinemia [6,66], and 1 (human) showing variable relationships between measures of metabolic syndrome and schistosome infection [69]. The latter, for instance, showed lower HOMA-IR and fasting insulin only in HIV-schistosome co-infected people not on ART, and not in those with HIV on ART or HIV-uninfected individuals with schistosome infection, when compared to normal controls [69]. This study also reported an increase in pancreatic beta cell function in those with schistosome infection who were HIV-uninfected.

Overall, studies investigating metabolic syndrome with non-schistosome helminth infections also reported similarly lower measures of metabolic syndrome-related parameters among infected individuals (n = 8) [16,17,47,58,59,63,65,69]. In contrast to the 22 total studies showing inverse associations between helminth infections and metabolic syndrome or parameters, 14 other studies showed no association [7,8,13,14,20,21,26,32,45,52,53,57,67,68]. These

14 studies included mostly human studies and investigated 4 *Schistosoma*, 4 mixed helminth, 3 *Strongyloides*, 1 *Opisthorchis*, 1 *Fasciola*, and 1 *Ascaris* infections.

**Summary**: In total, 38 studies reported on metabolic syndrome and related measures with a median sample size of 213.5 human and animal participants. Most (22 of 38, or 58%) demonstrated that helminth infections were associated with a lower frequency of metabolic syndrome, with most data coming from studies of *Schistosoma* infections. Of the 10 that reported on the effects of anthelmintic treatment, 7 showed that metabolic syndrome parameters increased after treatment of helminth infection.

When stratifying the human studies by age and sex, we conclude that in both men and women younger than 50 years, there was mostly no relationship (n = 9/15) versus lower measures of metabolic syndrome (n = 6/15) in association with helminth infections [7,8,13,21,26, 32,45,47,48,52,58,59,65,68,69]. On the other hand, in 5 of the 7 human studies [6,25,28,42,43, 53,56] with an average age of 50 or above, helminth infections were associated with lower prevalence of metabolic syndrome, suggesting that helminths may offer protection against metabolic syndrome in older individuals, particularly in men who were over-represented in these studies.

## Helminths and diabetes

Thirty studies investigated helminths and diabetes or related measures, such as hemoglobin A1c or pancreatic inflammation by histopathological analysis (S4 Table).

**Studies examining diabetes before and after anthelmintic treatment in humans.**   Five of these 30 investigated diabetes and related measures before and after anthelmintic treatment, all in humans [14,57–59,71]. The first of these followed up on a previous study [32] in which individuals with positive *S. stercoralis* serology had a significantly lower odds of prevalent type 2 diabetes (T2DM) [57]. Three years after testing and treatment for *S. stercoralis* infection, follow-up was successful in 80% of the original cohort participants (n = 207). Those who had been treated with ivermectin for *Strongyloides* infection had an unadjusted 7.7 times increased relative risk of being diagnosed with T2DM than the uninfected, untreated group [57]. After adjustment for age, sex, change in BMI and initial hemoglobin A1c, the relative risk for diabetes was lower but still trended towards significance (RR 5.85, 95% CI 0.75–39.5, p = 0.093).

In support of these findings, two prospective cohort studies also demonstrated worsened glycemic control after anthelmintic treatment. In *O. viverrini*-infected Thai individuals, who had a lower mean hemoglobin A1c (5.5%) than uninfected individuals (6.0%) at baseline, praziquantel treatment led to a significant increase in hemoglobin A1c (5.5%—>6.0%) over 6 months [14]. In a second study, individuals with T2DM and *Strongyloides* infection had similar hemoglobin A1c and random blood glucose levels, but lower insulin and glucagon levels, compared to uninfected individuals with T2DM at baseline [58]. Six months after albendazole and ivermectin treatment with confirmatory repeat parasitological examinations, previously infected individuals experienced increases in hemoglobin A1c by 25%, random blood glucose by 18%, and insulin level by 13% when compared to their pretreatment levels.

In contrast, another study led by the same first author found no effects of *Strongyloides* infection or anthelmintic treatment on hemoglobin A1c in non-diabetic, obese individuals 6 months after receiving albendazole and ivermectin [59]. Given a similar study design used by the same authors in these two studies [58,59], the different conclusions may reveal a preferential glycemic benefit of *Strongyloides* infection for people with known diabetes, and not for people without. Additionally, another study of children aged 9–14 years with mixed helminth infections in South Africa documented no differences in hemoglobin A1c at baseline or 6 months after treatment with albendazole [71]. However, potential selection bias and metabolic

differences between children and adults may limit this study's conclusions; it was also not reported whether the children were re-tested for infection or how many cleared their infection at follow-up.

No animal studies reported on diabetes and related measures before and after treatment of helminth infection.

In summary, 3 of 5 studies (60%; all human studies) that investigated markers of diabetes in people after anthelminthic treatment reported paradoxical worsening of diabetes and related measures after treatment [14,57,58]. Additional prospective data are warranted, particularly focusing on other helminth infections, clarifying post-treatment findings in children, and assessing whether age and sex affect the association between infection and diabetes.

**Cross-sectional associations of helminth infections with diabetes, glucose, or pancreatic inflammation.**   In total, 29 studies compared baseline measures of diabetes or pancreatic inflammation in those with and without helminth infection. Seventeen studies (59%; 11 animal, 6 human) reported lower measures of diabetes or pancreatic inflammation [14,25,32,68, 72–84], 6 (21%; all human) demonstrated higher measures of these factors [6,85–89], and 5 (17%; all human) showed no association with respect to helminth infections [53,58,59,71,90]. In the final study there was no difference in hemoglobin A1c or fasting glucose regardless of helminth infection or antiretroviral status among HIV-infected individuals, while among HIV-uninfected individuals, soil-transmitted helminth (STH) infection was inversely associated with hemoglobin A1c (5.2% vs. 5.5% in STH-uninfected controls) after adjustment for age and sex [69].

The 6 studies reporting higher measures of diabetes with helminth infections were all human studies with mixed helminths (2), *Schistosoma* species (2), and *Strongyloides* (2) represented. Among the 2 studies with mixed helminths, an increased odds of diabetes was reported for those with *Taenia* and *Ascaris* infections [86,87]. One case-control study among 300 people in Sudan found an increased odds of prevalent urogenital schistosome (aOR: 2.5, 95% CI: 0.8–7.8; p = 0.10) or intestinal parasite infection (aOR: 2.10, 95% CI: 0.97–4.5, p = 0.059) among diabetic participants, and the intensity of *S. haematobium* infection correlated with duration of T2DM (R = 0.666, p = 0.009) [85]. Similarly, the 5 studies showing no association were also all human studies, with 3 focused on *Strongyloides* infection. One study of 2867 Chinese participants with and without prior schistosome infection notably reported no difference in diabetes (19.2% vs. 22.4%, respectively; p = 0.063), although those with PSI had significantly lower lipids and CAD prevalence [53].

Of the 17 studies reporting lower measures of diabetes or pancreatic inflammation among helminth-infected participants, 6 involved humans [14,25,32,68,73,74]. In one study of 762 people, 2.6% of those with normal glucose tolerance, versus none of those with Type 1 diabetes, had positive antigen and antibody filarial tests for *W. bancrofti* and *B. malayi* [74]. A second study of 1416 participants demonstrated a dose-dependent inverse relationship between these filarial tests and T2DM, showing a lower prevalence of lymphatic filariasis among diabetic participants (both newly diagnosed [5.7%] and those under treatment [4.3%]) compared to pre-diabetic [9.1%, p = 0.0095] and non-diabetic participants [10.4%, p = 0.0463] [73]. Similar inverse relationships have also been described between those with PSI and diabetes in China [25], those with *O. viverrini* and hemoglobin A1c levels in Thailand [14], and those with prior *S. stercoralis* infection and diabetes among Aboriginal Australians [32].

In comparison to the 6 human studies, 11 animal studies with mouse models using *Schistosoma* species (n = 7), *Fasciola hepatica* (n = 2), and filarial species (n = 2) also documented lower measures of diabetes or pancreatic inflammation among helminth-infected animals [72,75–84]. Seven of 11 documented histopathological changes of the pancreas in mice and found that helminth infection was associated with less pancreatic inflammation, degradation,

and other architectural changes in mice who were infected or exposed to helminth antigens as compared to uninfected mice [72,76–81]. These studies also documented a lower incidence of diabetes between infected and uninfected mice, and via correlations with pancreatic histopathological changes suggested helminths may possibly protect against autoimmune diabetes. This potential insight into mechanistic causes of these inverse relationships is valuable given that human studies have been unable to examine for diabetes-related pancreatic changes.

**Summary**: In total, 30 studies reported on diabetes and related measures with a median sample size of 279.5 participants. Most (17 of 29, or 59%) demonstrated cross-sectional associations between helminth infections and lower frequency of diabetes or diabetes-related parameters. Infection with *Schistosoma* species were most represented in these studies. Of the 5 that reported on the effects of anthelmintic treatment, 3 (60%) showed that blood glucose increased after treatment of helminth infection. Among human studies that reported median age and sex distribution, 2 out of 5 [6,25,53,89,90] with median ages over 50 years observed a different cross-sectional trend and suggested that helminth infection may possibly increase diabetes risk in older women, although the sample size of studies is admittedly limited and women were over-represented in those 2 studies, which both focused on *Strongyloides* [89,90].

## Helminths and atherosclerotic cardiovascular disease

Eleven studies investigated the associations of helminths on CAD, MI, and/or large-vessel atherosclerosis. None investigated the subsequent effect of anthelmintic treatment on these outcomes (S5 Table).

**Cross-sectional associations between helminth infections and atherosclerotic cardiovascular disease in animals.**   Of the 6 mouse studies investigating effects of helminths on atherosclerosis [27,35,46,49,51,61], 5 showed fewer atherosclerotic plaques along the aorta and other arteries in schistosome-infected versus uninfected mice. One of these showed that *S. mansoni*-infected, ApoE-deficient mice had 50% less atherosclerosis in the aortic arch and brachiocephalic artery than uninfected mice [27]. Another reported similar findings, demonstrating that mice on HFD exposed to *S. mansoni* SEA had smaller aortic plaque size and 44% less progression of aortic root atherosclerosis versus mice on HFD exposed to phosphate buffered saline [49]. Additionally, a study investigating the effects of recombinant *S. japonicum* protein (rSj-Cys) showed that treatment with the protein significantly decreased atherosclerotic plaque along the entire aorta and aortic sinus, decreased fat deposition in the kidneys and glomerular damage, and improved blood flow to the heart in mice that were fed a HFD [51]. Two other studies [46,61] also found decreased aortic atherosclerosis in *S. mansoni*-infected mice, although the statistical significance was not clear in one of them. In contrast, a study in which mice received weekly *S. mansoni* egg exposure found that egg-exposed mice were not protected against the development or progression of atherosclerosis when fed a HFD, despite having lower total cholesterol and LDL, compared to unexposed mice on the same diet [35]. It should be noted that sex was not clearly described in the majority of these animal studies, that every animal study investigated the effects of schistosome infections, and that these studies examined only large-vessel atherosclerosis, not CAD or MI. Additionally, it should be noted that mouse studies are conducted in models with similar but altered physiology and under extreme conditions (e.g., genetically altered or fed extremely HFD) as compared to human studies.

**Cross-sectional associations between helminth infections and atherosclerotic cardiovascular disease in humans.**   Similar to the mouse studies, 1 human autopsy study also found that *Opisthorchis*-infected cadavers had significantly less aortic atherosclerosis than the autopsy cases without infection, and this was magnified with increasing infection burden

across all age ranges [38]. Only 12.2% of the cadavers were female. In contrast, two cross-sectional studies investigating the effects of mixed STH or filarial infections on carotid intimal thickness, a measure of carotid atherosclerotic vascular disease, found no association between infection and atherosclerotic plaque [22,47].

Only 3 studies reported on CAD or MI [22,53,91]. A cross-sectional study of 453 adults living in India showed no association between lymphatic filariasis and CAD [22]. Further, among those with CAD, there were no associations of lymphatic filariasis with ultrasound measurements of carotid intimal thickness, aligning with the mouse study mentioned above. In contrast, a second autopsy study also found less MI and atherosclerosis in autopsies with bilharzial cirrhosis compared to all other autopsies, but important possible confounders including age and sex of cadavers, and other types of cirrhosis, were not clearly described [91]. A third cross-sectional study of 2867 Chinese adults showed that prior schistosome infection was significantly associated with a lower prevalence of CAD, which remained significant even after adjusting for laboratory measures including serum lipids and hepatic dysfunction [53].

**Summary**: In total, 11 studies (6 animal, 5 human) reported on atherosclerotic cardiovascular disease with a median sample size of 319 study participants. None of these studies investigated the effects of anthelmintic treatment on these outcomes. Specifically, only 3 studies (all human) reported on CAD or MI [22,53,91], of which 2 (67%) found less CAD among those with schistosome infection. Of note, 7 of 10 studies (70%; 6 animal, 4 human) of large vessel atherosclerosis concluded that helminth infections were generally associated with less atherosclerosis, particularly in the aorta and carotid arteries [27,35,38,46,49,51,61]. Most demonstrating this inverse relationship were animal studies that investigated *Schistosoma* infections. When the human studies were stratified by age and sex, no notable trends were observed, though only a small number had clear reports of average age and proportion of women in their studies. Future studies should rigorously control for potential confounders, such as age, sex, physical activity, and other sociodemographic factors, that may be associated with both helminth infections and atherosclerotic cardiovascular disease.

## Helminths, blood pressure, and cardiovascular hemodynamics

Eighteen studies investigated relationships between helminth infection, blood pressure, and cardiovascular hemodynamics, such as peripheral vascular resistance and mean arterial pressure (S6 Table).

**Studies examining blood pressure and cardiovascular hemodynamics before and after anthelmintic treatment in humans.**   Three of the 18 studies investigated the effects of helminth infections on blood pressure before and after anthelmintic treatment, all in humans. One of these, a large prospective cohort study, found no differences in blood pressure before and 6 months after treatment of *O. viverrini* with praziquantel [14]. A second study was a cluster-randomized trial that demonstrated no differences in blood pressure before and after treatment of mixed helminth infections with albendazole at 52 weeks of follow-up [8]. The third, also a cluster-randomized trial, reported lower diastolic blood pressure in individuals with heavy *S. mansoni* infections compared to uninfected individuals at baseline [13]. However, follow-up blood pressures at 4 years were not affected by either quarterly or annual anthelmintic treatment with albendazole and praziquantel, although it should be noted that there were persistently high rates of infection in both treatment arms at follow-up. These studies mostly enrolled people younger than 50 years, limiting conclusions about older persons.

**Cross-sectional associations of helminth infections with blood pressure or cardiovascular hemodynamics.**   Together with the above 3 studies, 14 other human studies also compared baseline measures of blood pressure and cardiovascular hemodynamics in those with and

without helminths [7,23–25,32,42,43,47,48,53,56,68,91–92]. Of the studies investigating the associations of helminths with systolic blood pressure or hypertension, 13 human studies in a total of 18,807 people demonstrated no associations [7,8,13,14,24,25,32,42,43,47,48,53,68], 3 studies in a total of 781 people showed lower systolic blood pressure [23,56,91], and 1 reported higher systolic blood pressure in 213 pregnant women [92]. The three studies reporting inverse associations examined different helminth infections: *O. volvulus* (mean systolic blood pressure 105 mmHg vs. 113 mmHg in uninfected participants) [23], previous schistosome infection (mean systolic blood pressure 110 mmHg vs. 127 mmHg in those never infected) [56]; and autopsy cases with bilharzial cirrhosis (6.4% vs. 12.9% frequency of hypertension in all autopsy cases) [91]. In contrast, the single study with positive associations reported elevated systolic, diastolic, and mean arterial blood pressures in pregnant women with hookworm infection, and a 6-fold increased odds of elevated mean arterial pressure in those with *Trichuris*, of borderline significance [92]. However, the large majority of the studies, which included 95% of the human participants, indicated no associations of helminth infections with systolic blood pressure.

The same 17 human studies also investigated diastolic blood pressure with 11 showing no associations [7,8,14,23–25,32,47,48,53,68], 5 demonstrating lower diastolic blood pressure [13,42,43,56,91], and the same study of pregnant women showing higher diastolic blood pressure among infected participants [92]. Two of these studies, discussed in the prior paragraph, showed inverse associations between helminth infections and both systolic and diastolic blood pressure [56,91]. While helminth infection was not associated with systolic blood pressure, higher intensity of *S. mansoni* infection in a study of 1898 individuals from Uganda was associated with lower diastolic blood pressure (72.8 vs. 76.7 mmHg (heavy vs. light or moderate infection intensity, p = 0.01)) [13]. In further support of differential effects on diastolic versus systolic blood pressure, two cross-sectional studies of people with prior schistosome infection from China found lower diastolic pressure and less overall hypertension in men, but not women, and no differences in systolic blood pressure [42,43]. Notably, women were underrepresented in both of these studies. Among 11 studies showing no association between helminth infection and diastolic blood pressure, helminths represented included mixed species (3), *Schistosoma* species (3), *Strongyloides* (2), *Opisthorchis* species (2), and *O. volvulus* (1). Countries where these studies were conducted were also diverse and included Australia, China, Indonesia, Thailand, Ethiopia, Republic of Chad, and Uganda. Similar to the trend seen with systolic blood pressure, however, diastolic blood pressure seemed mostly to be unaffected by helminth infections with 11 of 17 studies showing no association between helminth infection and diastolic blood pressure. Interestingly, in the 4 studies with discordant systolic and diastolic blood pressure outcomes [13,23,42,43], 3 found that active or previous schistosome infection was associated with lower diastolic blood pressures and no impact in systolic blood pressures—perhaps suggesting differential effects on diastolic versus systolic blood pressures could be limited to schistosome infection.

Other measures of cardiovascular hemodynamics were reported in the single mouse study, which showed that mice infected with *S. mansoni* had lower peripheral vascular resistance and mean arterial pressure than uninfected mice [93].

**Summary**: In total, 18 studies (17 human, 1 animal) reported on blood pressure and cardiovascular hemodynamics with a median sample size of 555 study participants. The effects of helminths on blood pressure and peripheral vascular resistance were mixed, with no effect reported cross-sectionally in 10 (56%) of those studies. Additionally, of the 3 studies that reported on the effects of anthelmintic treatment, none showed that treatment of helminth infection had effects on follow-up blood pressure. When the human studies were stratified by age and sex, helminth infections were not associated with differences in blood pressure in either men or women under the age of 50. In contrast, among the 5 studies [25,42,43,53,56]

with older participants (median age 65.7 years [IQR 65.1–68.5] and 20.7% women [IQR 11.6%-27.7%]), helminth infection was consistently associated with lower diastolic blood pressure measurements. These data suggest that a possible beneficial effect of infection on diastolic blood pressure may occur more frequently in men over age 50 as compared to in women of the same age group.

## Helminths and hsCRP

Five studies examined helminth infections and hsCRP, an inflammatory marker that is used clinically to evaluate cardiac risk (S7 Table).

**Study examining hsCRP Before and after anthelmintic treatment in humans.** Only 1 of the 5 studies, all in humans, reported on the subsequent effects of anthelmintic treatment on hsCRP [8]. This double-blind, cluster-randomized, placebo-controlled trial reported hsCRP measurements at baseline and post-treatment among participants who received 4 rounds of albendazole or matching placebo every 3 months for 3 consecutive days. Infected individuals had one or more infections with *A. lumbricoides*, *T. trichiura*, *A. duodenale*, and *N. americanus*. The trial reported no difference in hsCRP between treatment groups at baseline or after 52 weeks of follow-up.

**Cross-sectional associations of helminth infections with hsCRP in humans.** Among the 5 studies of helminths and hsCRP, 2 in Indonesia reported no differences in hsCRP between those with mixed STH infections and those without [47,65]. In both studies, the geometric mean hsCRP was 0.5 mg/L or below in all groups, well below the hsCRP threshold of 2 or higher typically recognized to be associated with an increased risk for cardiovascular events [94,95].

In another cross-sectional study looking at CAD-positive vs. CAD-negative individuals living in India with positive versus negative filarial (*W. bancrofti* and *B. malayi*) antigen tests, the geometric mean hsCRP in the CAD-positive group was not different [22]. A fourth study in *S. haematobium*-infected vs. uninfected individuals in Gabon also found no difference in median hsCRP measurements [52].

**Summary**: In total, 5 studies reported on hsCRP with a median sample size of 646 participants, all in humans. None of the 5 studies showed any cross-sectional association between helminth infections and hsCRP, nor were any trends observed when the studies were examined by age and sex. The single study that reported the effects of anthelmintic treatment failed to demonstrate any effect of treatment of helminth infection on hsCRP.

## Helminths and non-atherosclerotic cardiovascular disease

Four studies examined the associations of helminths with non-atherosclerotic cardiovascular disease, defined in these studies as myocardial injury, myocardium lipid deposits, cardiac hypertrophy, eosinophilic myocarditis, or other endomyocardial manifestations of eosinophilia that may be related to helminth infections [46,96–98]. None of these studies investigated the subsequent effects of anthelmintic treatment on outcomes of non-atherosclerotic cardiovascular disease (S8 Table).

**Cross-sectional associations between helminth infections and non-atherosclerotic cardiovascular disease.** Of these 4 studies, 2 showed some increase in non-atherosclerotic cardiovascular disease [96,98], 1 reported less non-atherosclerotic cardiovascular disease [46], and 1 found no relationship between non-atherosclerotic cardiovascular disease [97] and helminth infections.

In a mouse model, acute *S. mansoni* infection was associated with abnormally lower density of cardiomyocytes, increased area of injury indicative of more myocarditis, and increased

collagen deposition on histopathological examination when compared to uninfected mice [98]. Further, as infection became more chronic, mice developed progressively worsening myocardial disease that was characterized by a further decrease in total cardiomyocytes. Notably, this study included only 20 mice, all of which were females. Another animal study reported an inverse relationship between *S. mansoni* infection and myocardium lipid deposits and cardiac hypertrophy in mice fed HFD or standard diet (SD) [46]. Investigators first showed that HFD versus SD led to abnormal myocardial deposits and cardiac hypertrophy on histopathology, then found that mice that were fed HFD and exposed to *S. mansoni* cercariae or soluble egg antigen developed fewer myocardial lipid deposits and cardiac hypertrophic changes than unexposed mice fed a HFD. Only male mice were used in this study.

In a third study, 50 young adult immigrants who had come from sub-Saharan Africa were stratified into those with mixed helminth infection and/or eosinophilia (study group) versus those without (control group) to evaluate for the presence of endomyocardial lesions [97]. The study group had thickened posterior mitral valve leaflets on both qualitative and quantitative echocardiography. No associations were observed between the type of helminth infection and valvular involvement or with degree of eosinophilia. Additionally, one other human study showed that *Opisthorchis* infection was associated with eosinophilic myocarditis in cadavers, but there were several limitations, including that it was not clear how infection was determined, whether a control group was used, or whether the reported outcomes reached statistical significance [96].

**Summary**: In total, 4 studies (2 animal, 2 human) reported on non-atherosclerotic cardiovascular disease with a median sample size of 45 study participants. The effects of helminths on non-atherosclerotic cardiovascular disease were mixed, with 2 studies showing more endomyocardial pathology, 1 demonstrating less, and 1 reporting no association between non-atherosclerotic cardiovascular disease and helminth infection. No studies investigated the effects of helminth eradication on non-atherosclerotic cardiovascular disease, which could provide helpful data to guide understanding of this area. Interestingly, the 2 mouse studies—both studying *S. mansoni*—had either all female or all male mice and came to different conclusions concerning the association between infection and cardiac changes. Further studies in this area are needed and should investigate whether sex and age may play a role in the relationship between helminth infections and non-atherosclerotic cardiovascular disease.

S9 Table summarizes the major findings for each category of cardiometabolic diseases and risk factors.

## Discussion

Our robust synthesis of 84 human and animal studies consistently demonstrates that helminth infections may protect against dyslipidemia, metabolic syndrome, diabetes, and atherosclerotic cardiovascular disease. This is particularly strengthened by the existence of longitudinal studies in which these parameters paradoxically increased after elimination of helminth infections. We posit that the ability of helminth infections to lower atherosclerotic cardiovascular disease risk seems to be mostly explained by a reduction in metabolic risk factors, such as dyslipidemia and diabetes, and may also be partly explained by the type of helminth infection. Our review confirms the findings of prior systematic reviews on metabolic outcomes [4,5] and extends them to examine post-treatment effects. Further, to our knowledge, this is the first systematic review to comprehensively evaluate the effects of a variety of helminth infections and anthelmintic treatments on multiple cardiometabolic diseases and risk factors. Importantly, these data show that the burden of cardiometabolic diseases and risk factors may paradoxically worsen as the World Health Organization and global community strive to eliminate human

helminthiases worldwide, which would have implications for at least 24% of the world's population [9,99].

Our data indicate that metabolic effects, rather than immune alterations, may explain the observed associations between helminth infections and lower cardiometabolic diseases. This is a notable finding given that helminth infections are well-known to induce type 2 and T regulatory immune responses while downregulating proinflammatory type 1 immune responses that are critical to the pathogenesis of some cardiometabolic diseases [99–104]. If lower cardiometabolic diseases were driven by helminth-induced immune changes, one might expect that hsCRP, which is elevated in cardiovascular disease and is used to stratify cardiac risk [94,95], would be lower in those with helminth infections [101,105,106]. Instead, we found no difference in hsCRP levels but strong, consistent differences in lipid and metabolic factors. We note that CRP is a nonspecific inflammatory marker [107] and we cannot conclude that lack of differences in CRP indicates lack of contribution of immune findings to our observations. However given the key role of the liver in lipid and glucose metabolism and the predilection of many helminths to affect this organ, we posit that altered intra-hepatic or other metabolic functioning may also contribute to the associations we have observed. One other important caveat is that we cannot exclude other immunologic mechanisms, including the liver's role in host immune responses [108], in analyzing the etiologies that may explain the inverse associations that we have reported.

Helminth infections are known to affect several metabolic pathways that could ultimately contribute to lowering cardiometabolic disease risk. For instance, it has been postulated that helminths could decrease serum lipids by parasitizing host dietary nutrients for their own survival, by altering the gut microbiota and thereby affecting intestinal lipid metabolism, or by regulating lipid metabolism in the liver or peripheral adipose tissue [52,109]. Helminths use host lipids for their own biosynthetic purposes, such as for membrane formation during larval stages and by incorporating fatty acids into parasite eggs [110]. An alternative hypothesis is that the lower adiposity in individuals with helminth infections could explain the inverse relationships observed between helminth infections with dyslipidemia and diabetes. Either of these hypotheses could be supported by our findings of little or no association between helminth infections and hypertension or non-atherosclerotic cardiovascular disease. One possible exception could be the association of helminth infections with lower diastolic blood pressure. A potential explanation for the lower diastolic blood pressure in people living with helminth infection could be arterial stiffness. Helminth infection may reduce diastolic blood pressure through either metabolic [111] or immunomodulatory effects [112–114]. The beneficial effect of helminth infection on lower diastolic blood pressure may be particularly profound in older adults [115], as 2 of the 3 positive studies for diastolic blood pressure enrolled participants with a mean age above 65 years [42,43].

Different helminths may have distinct effects on cardiometabolic diseases, potentially through direct physiologic or local metabolic effects on the liver. In support of this, over 90% (38 of 42) of studies with *Schistosoma* species, blood flukes that can affect the liver and intestines, demonstrated an association between infection and lower frequencies of cardiometabolic diseases and risk factors with infected participants generally having lower serum lipids, glucose, and other metabolic parameters than uninfected participants. In contrast, only 33% (4 of 12) of the studies with mixed helminths, 67% (6 of 9) of those with *Strongyloides*, and 75% (15 of 20) of those with all other helminths reported some inverse relationships between helminth infection and cardiometabolic risk. Given the smaller number of studies focused on other helminths compared to those that reported on *Schistosoma* species, future studies investigating the relationship between helminth infections and cardiometabolic diseases may consider focusing on these other helminths.

The mechanistic pathways that connect helminth infections with cardiometabolic diseases could provide new drug targets or other strategies to manage the global burgeoning of these cardiometabolic diseases. In particular, LDL was more consistently altered than HDL in helminth-infected participants, which may be similar to trends seen with the use of statin drugs [116]. This suggests that helminths may perhaps affect a specific aspect of the common pathway of lipoproteins [117] that preferentially alters LDL levels while relatively conserving HDL levels. Given the known clinical benefit of elevated HDL in lowering cardiovascular risk [18,19] and its inclusion in commonly used calculated risk equations (ASCVD 2013 Risk Calculator, Reynolds Risk Score, and the Framingham Risk Score with Lipids), several questions remain in our comprehensive literature assessment. First, do alterations in LDL lower cardiovascular risk in helminth-infected individuals chronically if HDL levels are largely unaffected? Second, given that LDL is more frequently altered in helminth infections, should calculation of cardiac risk in helminth-endemic areas be tailored to consider LDL, instead of HDL, levels?

Mechanistic clinical trials with translational components are needed to illuminate these pathways and answer these questions. Studies in which helminth infections are treated can incorporate measures of cardiometabolic diseases, which could guide future clinical screening and management for cardiometabolic diseases as we work towards global eradication of helminths. Prospective studies in humans must also account for crucial sociodemographic and clinical confounders that may be associated with both helminth infections and cardiometabolic diseases, which dramatically affected outcomes in some studies [42,52,69], particularly when women were under-enrolled. Examining trends seen in animal versus human studies further supports the need for rigorous control of potential confounding variables. Over 85% of animal studies reported less cardiometabolic diseases or risk factors in the setting of helminth infections, as compared to only ~64% of human studies, which suggests the presence of important confounders.

Our study has limitations. First, diagnostics used in the included studies vary by sensitivity and host characteristics, and in some studies were based on clinical history and subject to recall bias. Further, other studies did not document the clearance of helminth infections after treatment. These scenarios may lead to infections being underrepresented at baseline or overrepresented at follow-up, and may have influenced reported outcomes. Second, it is possible that medications could contribute to observed post-treatment effects. Praziquantel and albendazole, the two anthelmintics used in the majority of the included studies, do not have known direct effects on host glucose or lipid metabolism. However, ivermectin has been shown to decrease serum glucose and cholesterol levels in a few experiments [118,119]. Contrary to the results of those experiments, the 3 studies in this review that used ivermectin all reported increases in diabetes and/or metabolic syndrome, though the total sample sizes were small. Third, as with any systematic review, the risk of publication bias could have led to underreporting of studies that did not find associations between helminth infections and cardiometabolic diseases. Our systematic review was also limited to only the most prevalent human helminths, which may have excluded other important work examining the effects of non-human helminths on cardiometabolic diseases. Finally, due to the heterogeneity of the data with diverse study types, helminths, and cardiometabolic diseases and risk factors represented, a meta-analysis was not possible, which limits our ability to estimate overall effect sizes or to examine more comprehensively for the effects of confounders.

In summary, this rigorous, comprehensive systematic review demonstrates consistent interactions of helminths with dyslipidemia, diabetes, and atherosclerotic cardiovascular disease. Our data suggest that increased preventive care may be needed for individuals at moderate or high risk for cardiometabolic disease who receive mass drug administration for helminth infections. Providing these individuals with regular screening for diabetes, lipids, and

atherosclerotic cardiovascular disease could mitigate possible increased cardiometabolic risk after anthelmintic treatment in highly endemic areas, particularly those in which routine preventive primary care may not be available. Our findings highlight the need for further cross-disciplinary research, which would have implications for both individual and population health and could point towards ground-breaking new strategies to address the intersection between helminthiases and non-communicable diseases globally in regions where both commonly overlap.

## Supporting information

**S1 Table. Predefined search terms, keywords, and included study types.** (DOCX)

**S2 Table. Effects of Helminths and Anthelmintic Treatment on Lipid Profile (n = 45).** Abbreviations: IQR, interquartile range; STH, soil-transmitted helminths; HDL, high-density lipoprotein; TC, total cholesterol; LDL, low-density lipoprotein; trig, triglycerides; PCR, polymerase chain reaction; RCT, randomized controlled trial; PZQ, praziquantel; SEA, soluble egg antigen; CAD, coronary artery disease; PSI, previous schistosome infection; T2DM, type 2 diabetes; HFD, high-fat diet; BMI, body mass index. #study investigated other outcome measures that will be included in other tables. *denotes statistical significance, p<0.05. (DOCX)

**S3 Table. Effects of Helminths and Anthelmintic Treatment on Metabolic Syndrome (n = 38).** Abbreviations: IQR, interquartile range; STH, soil-transmitted helminths; PCR, polymerase chain reaction; BMI, body mass index; T2DM, type 2 diabetes; WC, waist circumference; PZQ, praziquantel; IR, insulin resistance; RCT, randomized controlled trial; HOMA-IR, homeostatic model assessment for insulin resistance; SEA, soluble egg antigen; PSI, previous schistosome infection; FBG, fasting blood glucose;; ART, antiretroviral therapy; HFD, high-fat diet; WHR, waist-hip ratio. #study investigated other outcome measures that will be included in other tables. *denotes statistical significance, p<0.05. (DOCX)

**S4 Table. Effects of Helminths and Anthelmintic Treatment on Diabetes (n = 30).** Abbreviations: IQR, interquartile range; STH, soil-transmitted helminths; PCR, polymerase chain reaction; T2DM, type 2 diabetes mellitus; HbA1c, hemoglobin A1c; PZQ, praziquantel; LF, lymphatic filariasis; T1DM, type 1 diabetes mellitus; PSI, previous schistosome infection; DM, diabetes mellitus; ART, antiretroviral therapy; STZ, streptozotocin; NOD, non-obese diabetic; SEA, soluble egg antigen; PBS, phosphate buffered saline; SWA, soluble extracts of worms. #study investigated other outcome measures that will be included in other tables. *denotes statistical significance, p<0.05. (DOCX)

**S5 Table. Effects of Helminths and Anthelmintic Treatment on CAD, MI, or Atherosclerosis (n = 11).** Abbreviations: CAD, coronary artery disease; MI, myocardial infarction; IQR, interquartile range; STH, soil-transmitted helminths; LF, lymphatic filariasis; PCR, polymerase chain reaction; PSI, previous schistosome infection; SEA, soluble egg antigen; HFD, high-fat diet; PBS, phosphate buffered saline. #study investigated other outcome measures that will be included in other tables. *denotes statistical significance, p<0.05. (DOCX)

**S6 Table. Effects of Helminths and Anthelmintic Treatment on Blood Pressure and Cardiovascular Hemodynamics (n = 18).** Abbreviations: IQR, interquartile range; STH, soil-

transmitted helminths; SBP, systolic blood pressure; DBP, diastolic blood pressure; PZQ, praziquantel; PCR, polymerase chain reaction; RCT, randomized controlled trial; BP, blood pressure; PSI, previous schistosome infection; MAP, mean arterial pressure; PVR, peripheral vascular resistance; #study investigated other outcome measures that will be included in other tables. *denotes statistical significance, p<0.05.
(DOCX)

**S7 Table. Effects of Helminths and Anthelmintic Treatment on High-sensitivity C-reactive Protein (n = 5).** Abbreviations: hsCRP, high-sensitivity C-reactive protein; IQR, interquartile range; STH, soil-transmitted helminths; PCR, polymerase chain reaction; RCT, randomized controlled trial; LF, lymphatic filariasis; CAD, coronary artery disease. #study investigated other outcome measures that will be included in other tables. *denotes statistical significance, p<0.05.
(DOCX)

**S8 Table. Effect of Helminths and Anthelmintic Treatment on Non-atherosclerotic Cardiovascular Disease (n = 4).** Abbreviations: IQR, interquartile range; STH, soil-transmitted helminths; HFD, high-fat diet; SD, standard diet; SEA, soluble egg antigen. #study investigated other outcome measures that will be included in other tables. *denotes statistical significance, p<0.05.
(DOCX)

**S9 Table. Trends of Cardiometabolic Diseases and Risk Factors Reported in All Included Studies.**
(DOCX)

**S1 Supplemental Table. Studies Excluded During Quality Assessment Using the Downs and Black Checklist.**
(DOCX)

**S1 PRISMA 2020 Checklist.** *From*: [120] **For more information, visit: http://www.prisma-statement.org/.**
(DOCX)

**S1 Appendix. Search Strategy for Systematic Review** *Search was rerun on March 2, 2022.
(DOCX)

**S2 Appendix. Extracted Data.**
(CSV)

## Acknowledgments

We thank the librarian team at Weill Cornell Medicine for their assistance with searching and retrieving study records for this systematic review. We also thank Drs. Mary Charlson, Carol Mancuso, and Martin Shapiro, who all lead the Master's Program in Clinical Epidemiology and Health Services Research at the Weill Cornell Graduate School of Medical Sciences, for their valuable input which helped to shape the direction of this study.

### Registration

A review protocol was registered with PROSPERO (registration number CRD42021228610). It is available online (https://www.crd.york.ac.uk/prospero/display_record.php?RecordID=228610).

## Author Contributions

**Conceptualization:** Khanh Pham, Keith Mages, Jennifer A. Downs.

**Data curation:** Khanh Pham, Anna Mertelsmann, Keith Mages, Jennifer A. Downs.

**Formal analysis:** Khanh Pham, Anna Mertelsmann, Jennifer A. Downs.

**Funding acquisition:** Khanh Pham, Jennifer A. Downs.

**Investigation:** Khanh Pham, Jennifer A. Downs.

**Methodology:** Khanh Pham, Keith Mages, Justin R. Kingery, Jennifer A. Downs.

**Project administration:** Keith Mages.

**Resources:** Keith Mages.

**Software:** Keith Mages.

**Supervision:** Robert N. Peck, Jennifer A. Downs.

**Validation:** Justin R. Kingery, Humphrey D. Mazigo, Hyasinta Jaka, Fredrick Kalokola, John M. Changalucha, Saidi Kapiga, Robert N. Peck, Jennifer A. Downs.

**Visualization:** Khanh Pham, Robert N. Peck, Jennifer A. Downs.

**Writing – original draft:** Khanh Pham, Jennifer A. Downs.

**Writing – review & editing:** Khanh Pham, Anna Mertelsmann, Keith Mages, Justin R. Kingery, Humphrey D. Mazigo, Hyasinta Jaka, Fredrick Kalokola, John M. Changalucha, Saidi Kapiga, Robert N. Peck, Jennifer A. Downs.

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
