## [Decision Letter · Decision Letter 0]

16 Sep 2022

Dear Pham,

Thank you very much for submitting your manuscript "Effects of Helminths and Anthelmintic Treatment on Cardiometabolic Diseases and Risk Factors: A Systematic Review" for consideration at PLOS Neglected Tropical Diseases. As with all papers reviewed by the journal, your manuscript was reviewed by members of the editorial board and by several independent reviewers. The reviewers appreciated the attention to an important topic. Based on the reviews, we are likely to accept this manuscript for publication, providing that you modify the manuscript according to the review recommendations. 

Reviewers have provided generally positive feedback. A couple of important suggestions are to make a more clear separation of human and animal studies, and to include some of the important relevant findings for worms outside of the top 10 most prevalent in human infections. Two different reviewers also suggest citing some existing related review papers, which would help provide some context for the novelty of this review.

Sincerely,

Bruce A. Rosa

Academic Editor

Ricardo Fujiwara

Section Editor

Reviewers have provided generally positive feedback. A couple of important suggestions are to make a more clear separation of human and animal studies, and to include some of the important relevant findings for worms outside of the top 10 most prevalent in human infections. Two different reviewers also suggest citing some existing related review papers, which would help provide some context for the novelty of this review.

Reviewer's Responses to Questions

**Key Review Criteria Required for Acceptance?**

**Methods**

-Are the objectives of the study clearly articulated with a clear testable hypothesis stated?

-Is the study design appropriate to address the stated objectives?

-Is the population clearly described and appropriate for the hypothesis being tested?

-Is the sample size sufficient to ensure adequate power to address the hypothesis being tested?

-Were correct statistical analysis used to support conclusions?

-Are there concerns about ethical or regulatory requirements being met?

Reviewer #1: (No Response)

Reviewer #2: No new analysis required; all well done. No Ethical issues.

Reviewer #3: Yes

**Results**

-Does the analysis presented match the analysis plan?

-Are the results clearly and completely presented?

-Are the figures (Tables, Images) of sufficient quality for clarity?

Reviewer #1: (No Response)

Reviewer #2: Yes

Reviewer #3: Yes

**Conclusions**

-Are the conclusions supported by the data presented?

-Are the limitations of analysis clearly described?

-Do the authors discuss how these data can be helpful to advance our understanding of the topic under study?

-Is public health relevance addressed?

Reviewer #1: (No Response)

Reviewer #2: Refer my general comment

Reviewer #3: Yes

**Editorial and Data Presentation Modifications?**

Reviewer #1: (No Response)

Reviewer #2: Minor revision

Reviewer #3: Please refer to comments attached.

**Summary and General Comments**

Reviewer #1: This manuscript aims to provide a systematic assessment of the causal relationship between helminth infection and cardiometabolic diseases and the effect of helminth eradication on

cardiometabolic risk. The analysis of 83 papers led to the conclusion that helminth infection may offer protection against dyslipidemia, metabolic syndrome, diabetes, and atherosclerotic cardiovascular disease, and that this protection may lessen after anthelmintic treatment.

A standard approach for the selection of manuscripts for the systematic review was adopted – however, it seems that only a small number of parasites were included in the search criteria. While this was justified by the authors through the claim that these were the top ten most prevalent in human infections, I think this is an important fact, which may have skewed the collection of research data. Therefore, I think this selection approach should be clearly stated in the abstract of the manuscript.

For each medical criteria a comprehensive comparison has been made with the effect of helminth infection. However, in terms of assessing the effect of helminth infection, I would prefer to see some delineation between the animal and human studies – these should not be grouped as a collective for the determination of efficacy of helminth infection. The animal studies are all experimental infections which typically use higher doses than are ever found in endemic human single infections. Furthermore, the infection progression is more defined in the animal studies, whereas the human population are deemed infected in this review if they present with a positive antigen test – although this does not mean that they are actually actively infected. 

The discussion was well considered and raised the interesting prospect that there may be non-immune effects of helminth infection which are mediating the protective effect that is seen in disease. 

However, I was surprised that the authors did not mention or refer to the recent systematic review which assessed the association of helminth infection with the metabolic syndrome and diabetes (doi: 10.3389/fendo.2021.728396.) – this should be considered in their discussion in comparison the findings presented here.

Reviewer #2: The study by Pham et al addressed a pertinent issue in their comprehensive systematic review which aimed to assess the reported effects of helminth infections and antihelminthic treatment on the development and/or severity of cardiometabolic diseases and risk factors. After analyzing 83 animal and human studies, the authors concluded that helminth infection may offer protection against dyslipidemia, metabolic syndrome, diabetes, and atherosclerotic cardiovascular disease. Overall, the manuscript is well written and can be further enriched by addressing the following comments:

1. Method: Clarify the statement "search was updated on March 1, 2022" versus data bases included until March 02, 2022.

2. Introduction: a bit shallow; better to show existing inconsistency in the literature though several studies favor inverse association between helminth infections and metabolic syndrome.

3. General questions on the discussion: Different helminths modulate the immune system differently and hence their effect varies. Besides, their effect could differ during acute and chronic infections with the typical example of schistosomiasis where infection with S mansoni for example induces production of both pro-inflammatory and anti-inflammatory as well as regulatory cytokines during acute and chronic infections, respectively. Did authors notice any difference between those studies which demonstrated inverse association versus those which didn’t in terms of chronicity of helminthic infections? Have authors noticed any common feature or limitation among those studies which reported the reverse or found no difference between infection and cholesterol level for example?

4. Authors suggested non-immunological mechanisms/other metabolic mechanisms could explain the protective role of helminth infections based on their observation of the lack of association with hsCRP levels. Although CRP is an acute phase reactant and marker of inflammation, will it be valid to exclude immunological mechanisms without looking into the other immunological markers? It is very true that one of the key roles of the liver is lipid and glucose metabolism; however, this same organ is the source of many of the cytokines including those that affect the metabolic functions of the liver. Given the cross-sectional nature of the studies, establishing causal relationship might be difficult and the metabolic mechanisms suggested by authors (which are not explained) have to be considered without excluding immunological causes

5. Finally, given the fact that WHO recommends periodic deworming to all at-risk people living in endemic areas, what will be the implications of the findings? Considering the protective role of helminthic infections can we recommend differently or where shall we strike the balance between keeping our old day worms and hence benefit from their co-existence versus their elimination (keeping in mind their bad side)?

6. Minor: please check reference listing format of the journal, some contain month. In addition, there are paragraphs/sentences throughout the document where reviewed studies’ findings are mentioned but with no references cited.

Other than these issues, this is an excellent comprehensive systematic review which revisits "the Hygiene hypothesis" with a different perspective.

Thank you

Reviewer #3: Please refer to comments attached.

PLOS authors have the option to publish the peer review history of their article (what does this mean?). If published, this will include your full peer review and any attached files.

Reviewer #1: No

Reviewer #2: Yes: Aster Tsegaye

Reviewer #3: No

Figure Files:

Data Requirements:

Reproducibility:

References

---

## [Editor Report · Decision Letter 1]

12 Dec 2022

Dear Pham,

We are pleased to inform you that your manuscript 'Effects of Helminths and Anthelmintic Treatment on Cardiometabolic Diseases and Risk Factors: A Systematic Review' has been provisionally accepted for publication in PLOS Neglected Tropical Diseases.

Best regards,

Bruce A. Rosa

Academic Editor

Ricardo Fujiwara

Section Editor

The authors have addressed all reviewer concerns. The clearer separation of human and animal studies, both in the text and in the tables has substantially improved the flow of the manuscript and it will now serve as a more useful resource for readers.

---

## [Editor Report · Acceptance letter]

3 Feb 2023

Dear Pham,

We are delighted to inform you that your manuscript, "Effects of Helminths and Anthelmintic Treatment on Cardiometabolic Diseases and Risk Factors: A Systematic Review," has been formally accepted for publication in PLOS Neglected Tropical Diseases.

Best regards,

Shaden Kamhawi

co-Editor-in-Chief

Paul Brindley

co-Editor-in-Chief
